# Current Situation and Perspectives on Hantaviruses in Mexico

**DOI:** 10.3390/v11070642

**Published:** 2019-07-12

**Authors:** Ana L. Vigueras-Galván, Andrés M. López-Pérez, Gabriel E. García-Peña, Oscar Rico-Chávez, Rosa E. Sarmiento-Silva, Gerardo Suzán

**Affiliations:** 1Laboratorio de Ecología de Enfermedades y Una Salud, Departamento de Etología, Fauna Silvestre y Animales de Laboratorio, Facultad de Medicina Veterinaria y Zootecnia, Universidad Nacional Autónoma de México, Av. Universidad 3000, UNAM-CU, Coyoacán and CDMX CP 04510, Mexico; 2Laboratorio de Virología, Departamento de Microbiología e Inmunología, Facultad de Medicina Veterinaria y Zootecnia, Universidad Nacional Autónoma de México. Av. Universidad 3000, UNAM-CU, Coyoacán, CDMX. CP 04510, Mexico

**Keywords:** hantavirus, Mexico, neglected disease, rodents, zoonoses

## Abstract

Hantaviruses are transmitted by rodents producing the hantavirus pulmonary syndrome (HPS) in the Americas. Today, no human cases of HPS have been reported in Mexico in spite of similar environmental conditions with Central America and the USA where several cases have occurred. To understand the current situation of hantaviruses in Mexico and the public health risk, a systematic review of studies was conducted reporting hantaviruses in rodents to known state seroprevalence and hantavirus genotypes. Simultaneously, this study identified the potential hantaviruses based on the phylogenetic diversity (PD) of hantaviruses reported in the Americas in hosts with the distribution in Mexico. A total 3862 rodents belonging to 82 species have been tested since 1999 to 2017. Overall, 392 individuals representing 43 rodent species were seropositive, and the seroprevalence ranged from 0 to 69.22%. Seven hantaviruses genotypes have been described in Mexico and three are zoonotic. Four host species of rodents are widely distributed in Mexico harboring the highest PD of viruses. According to the hosts distribution, 16 genotypes could be circulating in Mexico and some of these represent a potential risk for public health. This study proposed multidisciplinary and interinstitutional collaborations to implement systematic surveillance in rodents.

## 1. Introduction

Hantaviruses are RNA viruses of the genus *Orthohantavirus*, which are transmitted by rodents of the family Muridae. They cause hemorrhagic fever with renal syndrome (HFRS) and hantavirus pulmonary syndrome (HPS). These diseases are mainly associated with rural areas, as infection in humans occurs following the inhalation of aerosolized feces, urine, and saliva of infected rodents. Hemorrhagic fever with renal syndrome is reported in Europe and Asia and affects up to 30,000 people a year with a mortality rate of 0.1 to 15% [1]. Hantavirus pulmonary syndrome is endemic to the Americas and was first reported in 1993 in the four corners region in southwestern United States. When the first outbreaks started, 60% of infected people died and today, mortality decreased to 36%. In the United States, 731 cases were reported from 1993 to 2018, 40% of which were reported in the states along the United States-Mexico border [2,3,4]. Hantavirus pulmonary syndrome also occurs in different countries from Central America, such as Panama, where 259 human cases were reported from 1999 to 2016, with a mortality rate of 17% [5]. During 2018, a total of 103 confirmed cases of hantavirus have been reported in Panama. Further, 51 cases were classified as hantavirus fever without pulmonary syndrome and 48 cases were classified as HPS [6].

No human cases of HPS in Mexico have been reported to date in spite of several outbreaks of HPS reported in the USA southern border, and the constant reports of rodents harboring hantavirus antibodies along the northern border of Mexico. This disease is likely to occur in Mexico and probably has been misidentified due to the fact that signs and symptoms are similar to other prevalent diseases, such as dengue, leptospirosis, rickettsiosis, and influenza [7].

Reservoirs of hantaviruses responsible for HFRS in Europe and Asia are rodents of the family Muridae (subfamily Murinae) [8], while the reservoirs of HPS in the Americas are of the family Cricetidae (subfamilies Arvicolinae, Neotominae and Sigmodontinae) [9]. Hantaviruses have been occasionally reported in other rodent families, such as Heteromyidae and Echimyidae, as well as in bats, marsupials, and soricomorphs, but their potential as competent reservoirs is still unknown [10,11,12]. Thus far, more than 40 hantaviruses genotypes in the Americas have been reported in the scientific literature, of which at least 20 genotypes were considered zoonotic [13].

Mexico has similar environmental conditions with Central America (neotropical region), and with the southern states of the USA where several human cases of HPS have been reported. In addition, there are more than 120 species of cricetids in Mexico, which means a potentially large diversity of hosts and hantaviruses [14]. Based on this, the authors hypothesized that a great diversity of unknown hantaviruses were likely to occur all over the country. The objective of our study is to explore the situation in Mexico with regard to hantaviruses. To do this, the published reports of hantaviruses in Mexico were analyzed and the geographic distribution of the seroprevalence in wildlife hosts in the different states allover Mexico was examined. Based on an analysis of genomic sequences reported in GenBank, the genetic diversity of hantaviruses in all known reservoirs from the Americas was estimated to identify the rodent species currently present in Mexico that harbor the greatest genetic diversity of hantaviruses. With this information, the authors produced a map of the potential distribution of hantavirus reservoirs. In light of our results, guidelines and priorities for disease prevention and future research are suggested from both an ecological and public health perspectives.

## 2. Methods

The authors searched for reports in the ScienceDirect, Scopus and Web of Science databases using the search criteria (Hantavirus; Mexico; rodent host) and (Hantavirus; Mexico). In order to increase the number of national reports, the scientific literature in the TESIUNAM and INDIXIE databases of the Red Mexicana de Repositorios Institucionales (Mexican Network of Institutional Repositories) were searched, with the search criterion (Hantavirus). The authors found 27 documents published during the period from 1995 to 2017, and of these, 15 studies were used to identify the regional seroprevalence (Appendix A). The studies that did not identify species, geographic location or the number of individuals analyzed, as well as duplicate data, were excluded. One study reporting hantavirus antibodies in humans was also included, and unpublished data from three projects from the Laboratorio de Ecología de Enfermedades y Una Salud (Laboratory of Disease Ecology and One Health) of the FMVZ-UNAM. The studies were done in Hidalgo, Sonora and Mexico City [15,16,17].

In total, rodent studies were carried out in 65 localities distributed in 24 Mexican states out of 32. This study recorded a total of 82 rodent species analyzed belonging to four families: Cricetidae (65), Heteromyidae (14), Muridae (2) and Sciuridae (1).

Fourteen studies used enzyme-linked immunosorbent assay (ELISA) and one used immunofluorescence. The antigens used were Sin Nombre Virus (SNV), Caño Delgadito othohantavirus (CDV) and Montano *Orthohantavirus* (MTNV). Seroprevalence (*SP*) of hantavirus in rodents of Mexico was calculated as the number of individuals with hantavirus antibodies (*N_ip_*) divided by the total number of individuals analyzed (*N_i_*). The number of individuals analyzed by each species (Sp) varied from 1 to 414 and by State from 5 to 889. Thus, to reduce the variation in the estimates of seroprevalence due to the differences in the number of individuals analyzed by State and species, the seroprevalence was weighted (SP_w_) by the number of individuals analyzed (N_i_) log10 transformed [18,19]. In this way, the overrepresentation of species and States with a large sample size was avoided.
(1)SPw = log10 (Ni)×SP

To assess the genetic diversity of hantavirus circulating within each species, the authors calculated the phylogenetic diversity (PD) of 60 sequences from the S segment of the genome (encode the nucleocapsid protein, N) of hantaviruses reported in 41 reservoir species in America. Firstly, with the sequences, a phylogenetic tree using the neighbor-joining method by the Kimura two-parameter distance method was constructed. Secondly, the PD based on the phylogenetic tree was calculated by adding up the total branch length of the sequences analyzed per host species [20]. In this way, the host species that harbor the most diverse hantaviruses based on PD was identified.

The authors then determined whether or not these species were distributed in Mexico to establish the possible geographic distribution of reservoirs and orthohantaviruses in the country. To do this, maps were built of geographical distribution of the reservoir species currently present in Mexico that harbor the greatest genetic diversity of orthhantaviruses. The distribution layers available in the Geographic Metadata Catalog of the Comisión Nacional para el Conocimiento y uso de la Biodiversidad [21,22,23,24] and QGIS Geographical Information System were used.

## 3. Results

Rodent studies were carried out in 65 localities distributed in 24 Mexican states from 32. This study recorded a total of 82 rodent species analyzed belonging to four families: Cricetidae (65), Heteromyidae (14), Muridae (2) and Sciuridae (1). Fourteen studies used enzyme-linked immunosorbent assay (ELISA) and one used immunofluorescence. The antigens used were SNV, Caño Delgadito virus (CDV) and Montano virus (MTNV). The overall seroprevalence of hantavirus in rodents was 10.15% (392/3862). Rodent and *Orthohantavirus* surveillance have been conducted in 75% (24/32) of the Mexican states, of which 18 have reported antibodies against *Orthohantavirus* in rodents. Seroprevalence ranges from 0 to 69.22% among different regions of the country (Figure 1, Appendix A). Hidalgo (69.22%) had the highest seroprevalence, followed by Chihuahua (49.02%). The studies carried out in humans report a seroprevalence that ranged from 1.4 to 1.6% in the states of Colima, Hidalgo, Guanajuato, Mexico City, and Chihuahua [25].

Divided by rodent family, *Orthohantavirus* seroprevalences were 34.41% in Cricetidae, 34.01% in Heteromydae, 10.60% in Sciuridae and 10.43% in Muridae. This study recorded 43 seropositive species among the 82 analyzed. The seroprevalence by species ranged from 3% to 76% (Figure 2, Appendix A). Seropositive individuals were identified from the genera *Peromyscus*, *Reithrodontomys*, *Sigmodon*, *Oryzomys* and *Oligoryzomys*, which have species considered to be reservoirs of American orthohantaviruses. Seropositive individuals of the genera *Rattus* and *Mus* were also identified and considered reservoirs of Eurasian orthohantaviruses.

Thus far, seven *Orthohantaviruses* genotypes have been reported in 12 rodent species from ten Mexican states. These genotypes include the Montano orthohantavirus (MTNV), Huitzilac orthohantavirus (HUIV), Carrizal orthohantavirus (CARV) and Playa de Oro orthohantavirus (OROV), considered endemic to Mexico, the Limestone Canyon orthohantavirus (LSCV, not associated with illness) and two zoonotic hantaviruses, SNV and El Moro Canyon orthohantavirus (ELMC) (Table 1). In addition, given the distribution of potential rodent and soricomorphs hosts, at least 16 genotypes of hantaviruses previously reported in America could be present in Mexico (Table 2). From these 16 genotypes, at least nine are considered zoonotic, and seven have not still been associated with disease in humans. From the nine zoonotic viruses, six are from the USA and three are from Central America.

Twenty-two of the 41 species in which *Orthohantavirus* sequences have been reported in America are distributed in Mexico, and orthohantaviruses have been identified in Mexico in 14 of these species. According to the PD analysis, the rodent species *Oryzomys couesi*, *Reithrodontomys megalotis, Peromyscus maniculatus,* and *Peromyscus leucopus* harbor the highest hantaviruses PD, all of which are widely distributed in Mexico (Figure 3 and Figure 4).

## 4. Discussion

In Mexico, orthohantaviruses have been non-systematically monitored over the last two decades. Thus far, seven genotypes have been described and 82 host species have been identified. However, Mexico is one of the countries in Latin America with the lowest production of scientific literature about these viruses and their hosts. In the period of 1990–2016, only 2.41% of publications focused on hantavirus on the continent corresponding to Mexico, whereas Brazil (38.97%), Chile (23.38%) and Argentina (20.96%) produced the highest number of studies [26]. This reflects a lack of interest from both scientific and public health perspectives in Mexico, due to the apparent absence of HPS cases. Nevertheless, the similarity of ecosystems, wide distribution of host species, and geographic proximity to sites with HPS cases suggest the opposite.

In Mexico, seroprevalence of *Orthohantavirus* in rodents is 10.15%, which is higher than southwestern USA (6.9%), Colombia (2.1%–4.2%), and Panama (2.7%), but lower compared to Honduras (20.8%) [27]. In spite of the great variation in the sampling effort for all the states in Mexico, our analytical approach revealed seroprevalence trends. The highest seroprevalences were observed in the states of Hidalgo, Mexico City, and Mexico State. This might be explained by the distributions of reservoir species in the central region of the country, where at least four reservoir species with the highest genetic diversity of orthohantaviruses co-occur. There is regional variation in seroprevalence in the country, with up to 69.22% seroprevalence in the central region, 49.03% in the north, 14.88% in the southeast, and 19.89% in the west. This is consistent with the findings described by Miholland et al. (2017) who reported high variation in seroprevalence, explained mainly by differences in the species richness, composition, abundance of reservoir species, and habitat heterogeneity among regions [28].

The state of Chihuahua had the highest *Orthohantavirus* seroprevalence among rodents in the northern region. While there have been no reported cases of HPS in this state, there is evidence of seropositive humans [25]. Interestingly, it represents the area in the USA, along the Mexican border, where 299 cases of HPS were reported between 1993 and 2017 [3]. In a study carried out in humans, in the Yucatan state, there was a seroprevalence of 0.64%, while in rodents the seroprevalence at the same region was 14.9% [29]. In addition, there was a seroprevalence in humans between 1.4% to 1.6% in the states of Colima, Hidalgo, Guanajuato, Mexico City, and Chihuahua [25]. Consistently, all of those states have had evidence of seropositive rodents except Guanajuato (Figure 1), as well as molecular evidence of OROV and LSCV genotypes in Colima and Mexico State, respectively.

There are 254 species of rodents distributed among nine families in Mexico [30]. Our findings provide evidence that the orthohantaviruses have been in contact with the most abundant and widely distributed families in the country—Muridae, Cricetidae, Heteromydae and Sciuridae—with high seroprevalences. Six genera that comprise rodent species recognized as orthohantaviruses reservoirs in America are present in Mexico: *Peromyscus*, *Reithrodontomys*, *Sigmodon*, *Oryzomys*, *Oligoryzomys* and *Microtus*. Among these genera, only *Microtus* did not show seropositive individuals in Mexico. The species of the genus *Peromyscus* are the most represented among the samples, and harbor the highest seroprevalence among the species of cricetids in Mexico. The species of the genus *Peromyscus* are considered the main reservoirs of hantavirus in North America, along with the genera *Reithrodontomys* and *Sigmodon*, while the genera *Oligoryzomys* and *Oryzomys* are potential hantavirus reservoirs in Central and South America.

Six species of the family Heteromyidae have been reported with seropositive individuals in Mexico, particularly the genus *Dipodomys* and the species *Liomys irruratus*, which is a generalist species. Seropositivity among heteromyds is often discarded and considered an accidental [31] result. However, the heteromyds seem to be directly related to the prevalence and maintenance of orthhantavirus in rodent communities, even though their role as reservoirs is unknown [28].

This study identified seropositive individuals of the family Muridae (*Rattus rattus* and *Mus musculus*) in Hidalgo, Veracruz and Yucatán. These species are widely distributed in the country and are considered among the most damaging invasive species for ecosystems [32]. Although these findings could be the result of cross-reaction with other orthohantaviruses, systematic surveillance in these synantropic species should be implemented. In Chile, both serological and molecular evidence of the Andes Virus have been found in the species *Rattus novergicus* and *Rattus rattus* [33,34]. Recently, the first known transmission of the Seoul virus (SEOV) from rats (*Rattus novergicus*) to humans was reported in the United States and Canada [2]. These findings demonstrate the importance of murid rodents in the transmission and maintenance of viruses among rodents in urban and suburban areas.

Two zoonotic orthhantaviruses have been identified in rodents in Mexico distributed among different states in the country in which no HPS or seropositive humans have been reported (Figure 1, Table 2). The viruses MTNV, HUIV and CARV, considered endemic to Mexico, are phylogenetically closely related to ELMC and LSCV [35], while the OROV is closely related to Bayou virus (BAYV) and Catacamas virus (CATV), but is suggested to be a unique genotype [36]. The 16 orthhantaviruses potentially present in Mexico could be circulating among populations of rodents, as well as soricomorphs. These mammals, in which these hantaviruses have been reported, are distributed in Mexico [11,12,37].

Our finding of high phylogenetic diversity of hantaviruses may be explained by the wide distribution of rodent species, potentially as enzootic reservoirs of *Orthohantavirus*, in Mexico This could also be explained by the biogeographic convergence that makes Mexico a region with high biodiversity, including high pathogen diversity. Another explanation is that habitat loss and fragmentation of native ecosystems is associated with increases in the abundance of *Orthohantavirus* reservoirs, and decreases in the species richness of non-host small mammal species [38]. Further research on the diversity of pathogens is important in disease ecology because it allows recognizing ecological associations and evolutionary trends between pathogens and hosts.

Further topics that should be considered for future research include the identification of host switching events and the role of species co-occurrence in infection dynamics. The transmission of orthohantaviruses between species is recognized to favor diversification of these viruses [39]. In Mexico, orthohantaviruses are circulating in different families, genus and species of rodents, and this maybe the result of host switching events [10]. Similarly, studies describing species co-occurrence data have identified novel competent reservoirs in different systems [40,41]. This approach may help to identify potential *Orthohantavirus* reservoirs in Mexico and recognized areas of high public risk for HPS cases. While substantial knowledge has been accumulated on hantavirus in Mexico, the establishment of disease surveillance programs and the ability to identify cases of hantavirus continue to be strong challenges in the country. This may be possible with the multidisciplinary and inter-institutional participation of the different groups involved in the scientific and public health sectors.

## 5. Conclusions

Hantavirus pulmonary syndrome is a neglected disease in Mexico. Therefore, it is important to strengthen and improve diagnostic strategies in first contact healthcare and in the reference laboratories of the health system for timely detection, establishing appropriate therapies and finally, to know the epidemiological situation of the disease in Mexico. The investigation of the wide diversity of hosts and the enzootic cycles of *Orthohantavirus* is key in the surveillance of this pathogen.

## Figures and Tables

**Figure 1 viruses-11-00642-f001:**
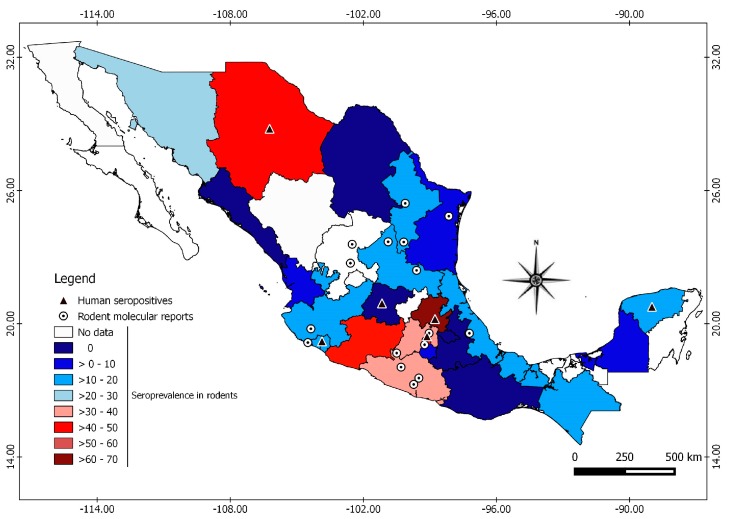
Geographic distribution of rodent and human reports of orthohantaviruses in Mexico. The seroprevalence in rodents is shown in the colored states, the molecular evidence in rodents with circles, and the serological evidence in humans with triangles.

**Figure 2 viruses-11-00642-f002:**
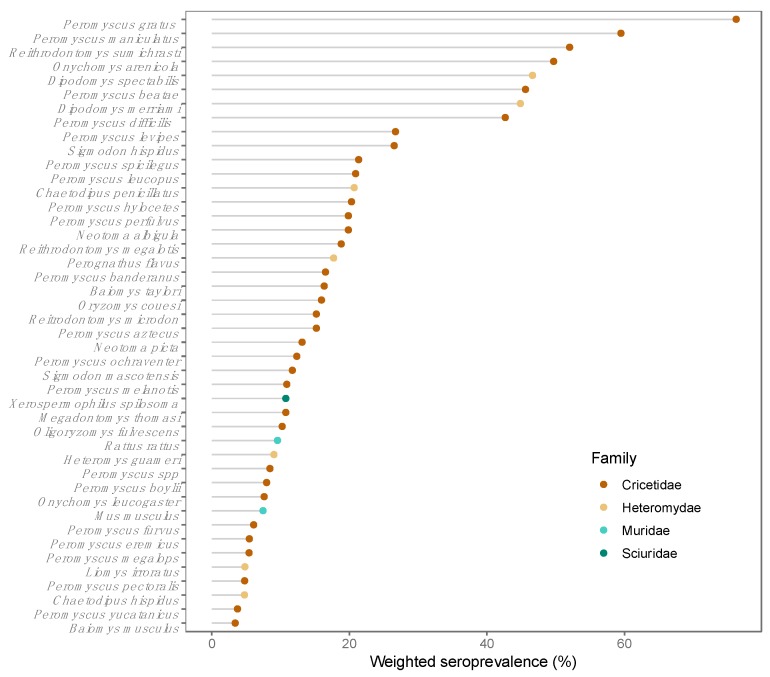
Seropositive species and weighted prevalence.

**Figure 3 viruses-11-00642-f003:**
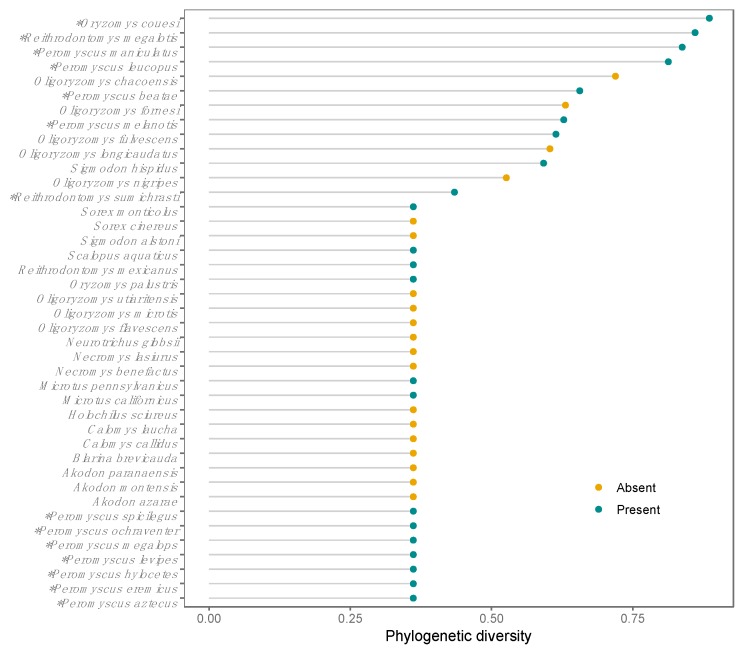
Phylogentic diversity of *Orthohantavirus* per reservoir in the Americas. Coloured points showed species absent and present in Mexico. * Species in which orthohantaviruses genotypes have been identified in Mexico.

**Figure 4 viruses-11-00642-f004:**
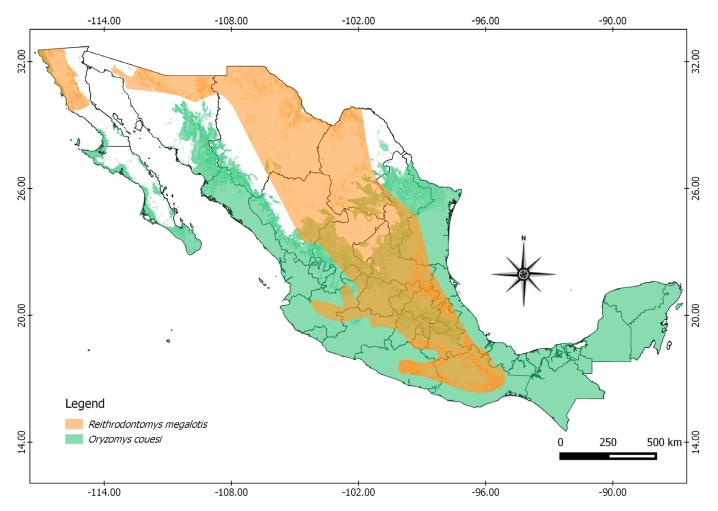
Geographic distribution in Mexico of reservoirs harboring highest *Orthohantavirus* phylogenetic diversity.

**Table 1 viruses-11-00642-t001:** Genotypes of orthohantaviruses identified in Mexico by species and State.

State	Species	*Orthohantavirus* Genotype
Zacatecas	*Reithrodontomys megalotis*	ELMC *
	*Reithrodontomys megalotis*	SNV *
Colima	*Oryzomys couesi*	OROV
	*Sigmodon mascotensis*	OROV
Guerrero	*Peromyscus batae*	MTNV
	*Peromyscus aztecus*	MTNV
	*Reithrodontomys sumichrasti*	CARV
	*Peromyscus megalops*	CARV
Morelos	*Reithrodontomys megalotis*	HUIV
Guerrero	*Peromyscus batae*	HUIV
Guerrero	*Reithrodontomys sumichrasti*	ELMC *
Jalisco	*Peromyscus spicilegus*	LSCV
Mexico State	*Peromyscus melanotis*	LSCV
	*Peromyscus hylocetes*	LSCV
Nuevo León	*Peromyscus maniculatus*	SNV *
	*Peromyscus eremicus*	SNV *
	*Peromyscus levipes*	LSCV
San Luis Potosí	*Peromyscus maniculatus*	SNV *
	*Peromyscus ochraventer*	LSCV
Tamaulipas	*Peromyscus leucopus*	SNV
Veracruz	*Reithrodontomys megalotis*	ELMC
	*Peromyscus melanotis*	ELMC
	*Peromyscus maniculatus*	SNV

* Zoonotic.

**Table 2 viruses-11-00642-t002:** Reservoirs present in Mexico and their orthohantaviruses genotypes.

Reservoir	Genotypes Orthohantaviruses	Country	Disease
*Oligoryzomys fulvescens*	Choclo orthohantavirus	Panama	HPS
*Oryzomys palustris*	Bayou orthohantavirus	USA	HPS
*Sigmodon hispidus*	Black Creek Canal orthohantavirus	USA	HPS
*Peromyscus maniculatus*	Monongahela orthohantavirus	USA	HPS
*Peromyscus leucopus*	New York orthohantavirus	USA	HPS
*Microtus pennsylvanicus*	Prospect Hill orthohantavirus	USA	NR
*Oligoryzomys fulvescens*	Maporal orthohantavirus	Venezuela	NR
*Oryzomys couesi*	Catacamas orthohantavirus	Honduras	HPS
*Oryzomys couesi*	Caño delgadito orthohantavirus	Honduras	HPS
*Sorex monticolus*	Jemez Springs orthohantavirus	USA	NR
*Scalopus aquaticus*	Rockport orthohantavirus	USA	NR
*Microtus californicus*	Isla Vista orthohantavirus	USA	NR
*Peromyscus leucopus*	Blue River orthohantavirus	USA	HPS
*Reithrodontomys mexicanus*	Rio Segundo orthohantavirus	Costa Rica	NR
*Sigmodon hispidus*	Muleshoe orthohantavirus	USA	NR
*Peromyscus maniculatus*	Convict Creek orthohantavirus	USA	HPS

HPS: Hantavirus Pulmonary Syndrome; NR: Not recognized.

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
