# Peer review of "Current Situation and Perspectives on Hantaviruses in Mexico"

_viruses, 2019, doi:10.3390/v11070642_

Round 1
Reviewer 1 Report
The manuscript by Vigueras-Galván et al reviews the evidence available for the incidence of hantaviruses in reservoir hosts and an apparent lack of human hantavirus cases in Mexico. The manuscript addresses a key question in the hantavirus field and is pertinent. The authors collate and analyze available data from previous hantavirus studies in Mexico and present a scenario that supports the likely scenario of under-diagnosis of hantavirus infections and suggest ways to further strengthen surveillance.
Specific points:
Authors calculate weighted seroprevalence and phylogenetic diversity to compare different studies. Authors should explain these calculations in more detail in the methods section and potentially include examples of some calculations.
Although the authors have done pretty good job writing the manuscript, multiple places in the manuscript can be polished further for English language. Below are some examples (but this is not an exhaustive list):
line 29 - "in spite similar" should be "in spite of similar"
Table 1 - line 3 - should "ELMCV" be "EMCV"?
line 132 - "exposure" should be "exposed".
lines 141-142 - "have the highest production of studies" could be revised to "produced the highest number of studies".
line 179 - "heteromids" should be "hetermyds"
Author Response
Reviewer 1
The manuscript by Vigueras-Galván et al reviews the evidence available for the incidence of hantaviruses in reservoir hosts and an apparent lack of human hantavirus cases in Mexico. The manuscript addresses a key question in the hantavirus field and is pertinent. The authors collate and analyze available data from previous hantavirus studies in Mexico and present a scenario that supports the likely scenario of under-diagnosis of hantavirus infections and suggest ways to further strengthen surveillance.
We appreciate your comment about our work
Point 1. Authors calculate weighted seroprevalence and phylogenetic diversity to compare different studies. Authors should explain these calculations in more detail in the methods section and potentially include examples of some calculations.
Response 1. We have made the suggested changes in the methodology clarifying weighted seroprevalence and phylogenetic diversity analyses. See lines 91-108
Although the authors have done pretty good job writing the manuscript, multiple places in the manuscript can be polished further for English language. Below are some examples (but this is not an exhaustive list)
Point 2. line 29 - "in spite similar" should be "in spite of similar"
Response 2. Done as suggested
Point 3. Table 1 - line 3 - should "ELMCV" be "EMCV"?
Response 3. We have checked the acronym accepted by the ICTV. The correct acronym should be ELMC
Point 3. line 132 - "exposure" should be "exposed"
Response 3. Done as suggested
Point 4. lines 141-142 - "have the highest production of studies" could be revised to "produced the highest number of studies"
Response 4. Corrected.
Point 5. line 179 - "heteromids" should be "hetermyds
Response 5. We have checked should be heteromyds
Reviewer 2 Report
Review of Viruses-518185 entitled “Current situation and perspectives on hantaviruses in Mexico” for potential publication in Viruses MDPI.
Ana L. Vigueras-Galván and colleagues attempt to address hantavirus distribution and seroprevalence across Mexico and for different host species, and plan to discuss their results in light of the impact for public health. I think the study is valuable for Viruses MDPI readers, and an interesting contribution in the area that link viral zoonotic pathogens, ecology and public health. However, there are several ideas that need to be corrected and/or clarified before the manuscript is published. I detail those points as follow:
Line 53. Add reference
Line 59. The authors should clarify if they refer to genotypes, strains, or lineages.
Lines 59-60 and 118-122 and tables. In addition to genotypes/lineages, how those fit within the current ICTV nomenclature?
Lines 65-66. It is not clear what the authors mean with the sentence “In order to better understand the hantaviruses situation in Mexico….”.
Lines 68-69. What do they mean with: “potential genetic diversity”?
Lines 85-88. The analysis of weighted seroprevalence is interesting. Also, the authors should detail the effort of sampling or discuss the limitations of their values. It is known that increasing the sampling effort may result in higher chances to capture seropositives.
Lines 89-93. Due to inferences associated with PD, a more detailed methodology should be added.
Lines 99-101. The sentence should be within Methods section, and the limitations of the differences should be explained.
Figure 1. It is an informative figure, but the gray scale should be improved to better visualization (e.g. additional or alternative texture).
Line 132. “….have been exposure to hantaviruses…”. What do they mean?
Figure 3A. Y-axis is not clear due to overlapping.
Figure 3B. It is not clear in the Methods section how the map was built using the potential PD analyses and the distribution of the hosts.
Lines 141-142. The sentences need some work to make them completely clear.
Line 149. Probably a word is missing to make the sentence clearer.
Lines 181-182. Rattus species are known to harbor other hantavirus strains different to those from the Americas. How the seropositves in Rattus species reported in this study may be biased due to the methods for detecting those antibodies?
Lines 183-185. Earlier, Lobos et al (2005) also reported in Chile positives for ANDV in Rattus.
Lines 197-204. I believe that the authors can improve this part of the discussion to strength the MS. They include distribution of host species, convergence that increases the diversity, habitat lost and fragmentation, but there are additional factors that may explain changes in hantavirus dynamics. For example, what is the role of host switch? How the factors they describe plus others such as host switch are related to the results they report? How the nice results they collected are important to understand the ecology of the hosts and epidemiology?
Lines 199-200. Also, in light of their results, how high pathogen diversity is critical for understanding disease ecology? Additional sentences will improve the discussion.
Author Response
Ana L. Vigueras-Galván and colleagues attempt to address hantavirus distribution and seroprevalence across Mexico and for different host species, and plan to discuss their results in light of the impact for public health. I think the study is valuable for Viruses MDPI readers, and an interesting contribution in the area that link viral zoonotic pathogens, ecology and public health. However, there are several ideas that need to be corrected and/or clarified before the manuscript is published. I detail those points as follow.
We appreciate your comment about our work
Point 1. Line 53. Add reference
Response 1. The reference is included. See line 53
Point 2. Line 59. The authors should clarify if they refer to genotypes, strains, or lineages
Response 2. In the sentence, we mean “genotypes”. See lines 59-60
Point 3. Lines 59-60 and 118-122 and tables. In addition to genotypes/lineages, how those fit within the current ICTV nomenclature?
Response 3. We have made the modification in the text according to the new ICTV nomenclature for orthohantavirus (previously hantavirus). Corrected
Point 4. Lines 65-66. It is not clear what the authors mean with the sentence “In order to better understand the hantaviruses situation in Mexico….”
Response 4. We corrected this sentence clarifying the objective of the study. See lines 65-68
Point 5. Lines 68-69. What do they mean with: “potential genetic diversity”
Response 5. Based on an analysis of genomic sequences reported in GenBank, we estimate the genetic diversity of hantaviruses in all known reservoirs from the Americas to identify the rodent species currently present in Mexico that may harbor the greatest genetic diversity of hantaviruses. With this information we produce a map of potential distribution of hantaviruses reservoirs that harbor the highest genetic diversity of hantaviruses. See lines 68-72
Point 6. Lines 85-88. The analysis of weighted seroprevalence is interesting. Also, the authors should detail the effort of sampling or discuss the limitations of their values. It is known that increasing the sampling effort may result in higher chances to capture seropositives.
Response 6. The weighted seroprevalence was calculated because the sampling effort was different throughout the localities. To reduce the variation in the estimates of seroprevalence due to the differences in the number of individuals analyzed by State and species, the seroprevalence was weighted by the number of individuals analyzed. See lines 93-100
We specified this in the methods and in the discussion sections see lines 171-172
Point 7. Lines 89-93. Due to inferences associated with PD, a more detailed methodology should be added
Response 7. Done as suggested. See lines 102-108
Point 8. Lines 99-101. The sentence should be within Methods section, and the limitations of the differences should be explained.
Response 8. Done as suggested. See lines 86-88
Point 9. Figure 1. It is an informative figure, but the gray scale should be improved to better visualization (e.g. additional or alternative texture)
Response 9. Corrected as suggested and the quality of the figure was improved
Point 10. Line 132. “….have been exposure to hantaviruses…”. What do they mean?
Response 10. We have corrected the sentence for a better understanding. See lines 151-155
Point 11. Figure 3A. Y-axis is not clear due to overlapping
Response 11. We have improved the quality of the figures. See figure 3
Point 12. Figure 3B. It is not clear in the Methods section how the map was built using the potential PD analyses and the distribution of the hosts
Response 12. We have specified the methodology to create the maps. See lines 109-114
Point 13. Lines 141-142. The sentences need some work to make them completely clear
Response 13. We have made the correction of the sentence for better understanding. See lines 161-165
Point 14. Line 149. Probably a word is missing to make the sentence clearer
Response 14. We have made the correction of the sentence for better understanding. See lines 172-175
Point 15. Lines 181-182. Rattus species are known to harbor other hantavirus strains different to those from the Americas. How the seropositves in Rattus species reported in this study may be biased due to the methods for detecting those antibodies?
Response 15. Rattus spp has been identified with Euroasian hantaviruses like Puumala and Seoul viruses, but also with other hantaviruses from the Americas like Andes virus in Chile and the Seoul virus in the United States and Canada. In Mexico Rattus spp have been reported with antibodies against hantavirus using SNV virus antigens. This may be a result of a cross-reaction with other hantaviruses like Andes and Pummala. In this case, we can not specify which hantavirus genotype is circulating. For this reason, we consider that further molecular analyses are needed. See lines 209--211
Point 16. Lines 183-185. Earlier, Lobos et al (2005) also reported in Chile positives for ANDV in Rattus
Response 16. Thank you for this clarification, we have included this reference in the text. See lines 212
Point 17. Lines 197-204. I believe that the authors can improve this part of the discussion to strength the MS. They include distribution of host species, convergence that increases the diversity, habitat loss and fragmentation, but there are additional factors that may explain changes in hantavirus dynamics. For example, what is the role of host switch? How the factors they describe plus others such as host switch are related to the results they report? How the nice results they collected are important to understand the ecology of the hosts and epidemiology?
Response 17. We have improved the discussion. See lines 233-237
Point 18. Lines 199-200. Also, in light of their results, how high pathogen diversity is critical for understanding disease ecology? Additional sentences will improve the discussion
Response 18. We have improved the discussion. See lines 235-240
Reviewer 3 Report
The authors report on the topic of Hantaviruses in Mexico. Very little is known about hantavirus in Mexico and it is very refreshing to see this topic covered by the authors and should be of interest to readers. This article may be presented more appropriately as a Review rather than an Article given that the majority of data is mined from the scientific literature.
Minor Comments:
-There were several minor grammatical errors as well as awkward sentences that could be improved through out the manuscript. I have provided a list of the locations in the manuscript:
Lines: 31-32, 42-43, 59, 63, 97, 103, 106, 139, 201
Major Comments:
-The methods make mention of unpublished data. Its not clear in the manuscript and analysis which data came from published work and/or unpublished work. In addition, I presume the unpublished data was acquired by the authors. If so, the full details of the methods should be included (ie. ELISA methods, antigens used, rodent trapping methods etc.)
-It would be beneficial to the readers to identify what data came from which reference in the figures and tables presented, as well as data that was unpublished.
-The figures and tables are lacking appropriate titles and captions. Some figures have no captions at all describing the data presented. This will need to be addressed in the final version.
Author Response
The authors report on the topic of Hantaviruses in Mexico. Very little is known about hantavirus in Mexico and it is very refreshing to see this topic covered by the authors and should be of interest to readers. This article may be presented more appropriately as a Review rather than an Article given that the majority of data is mined from the scientific literature
We appreciate your comment about our work
Point 1. There were several minor grammatical errors as well as awkward sentences that could be improved through out the manuscript. I have provided a list of the locations in the manuscript:
Lines: 31-32, 42-43, 59, 63, 97, 103, 106, 139, 201
Response 1. Corrected
Point 2. The methods make mention of unpublished data. Its not clear in the manuscript and analysis which data came from published work and/or unpublished work. In addition, I presume the unpublished data was acquired by the authors. If so, the full details of the methods should be included (ie. ELISA methods, antigens used, rodent trapping methods etc.)
Response 2. We have reviewed and restructured the methodology for a better understanding. Additionally, we included a supplementary table S1 in which we specified the diagnostic technique and the antigen used by each study, including the methodology used for the unpublished studies. See supplementary table S1
Point 3. It would be beneficial to the readers to identify what data came from which reference in the figures and tables presented, as well as data that was unpublished.
Response 3. We have included additional information in the supplementary table to clarify this. See supplementary table S1
Point 4. The figures and tables are lacking appropriate titles and captions. Some figures have no captions at all describing the data presented. This will need to be addressed in the final version
Response 4. We have corrected the figures and tables including titles and captions